# Examining Technology Acceptance in Learning and Teaching at a Historically Disadvantaged University in South Africa through the Technology Acceptance Model

**Clever Ndebele** [1,*] **and Munienge Mbodila** [2]

[1] Directorate of Learning and Teaching , Walter Sisulu University, Mthatha 5117, South Africa
[2] Department of Information Technology Systems, Walter Sisulu University, Komani Campus, East London 5200, South Africa; mmbodila@wsu.ac.za
\* Correspondence: cndebele@wsu.ac.za

**Abstract:** The exponential growth in the use of technology for learning and teaching in the higher education sector has imposed pressure on academics to embrace technology in their teaching. The present study sought to examine factors underlying technology acceptance in learning and teaching at a historically disadvantaged university in the Eastern Cape Province of South Africa. Premised on the mixed methods approach and undergirded by the Technology Acceptance Model (TAM), both a pre-coded and an open-ended questionnaire were used to collect data. Data from the pre-coded questionnaire were analysed through the descriptive statistical approach. The qualitative data from the open-ended questionnaire were analysed through content analysis. The study found that most academic staff believe and see the value that ICTs bring in their teaching and learning practices. In addition, they are aware that technology use in education improves learning and teaching, and they are willing to embrace the use of technology to improve their practices. Based on the findings, we recommend intensification of lecturer training in the use of technology for teaching and learning to enable them to embrace it in their teaching practice. Furthermore, the institution needs to put in place support systems for academic staff to empower them to have continuous access to devices and internet connection for technology integration in teaching and learning. We recommend establishment of e-learning communities of practise in the university that will allow lecturers to assist each other as well as share best practices in the use of technology for teaching and learning.

**Keywords:** e-learning; technology acceptance; learning management system; behavioral intention e-learning; technology acceptance; learning management system; behavioral intention





## 1. Introduction

The exponential growth in the use of technology for learning and teaching in the higher education sector has imposed pressure on academics to embrace this technology in their teaching. In South Africa, in 2015, with the onset of the '#FeesMustFallMovement' in universities, even more pressure has mounted to embrace technology in learning and teaching during times of disruption. Across the system, as Czerniewicz, Trotter and Haupt [1] show, university leadership engaged to varying degrees with protestors' demands, while simultaneously considering and using measures that would allow teaching to continue, or at least for the curriculum to be completed, to circumvent the effects of the disruptions with blended learning emerging as one of these measures.

The outbreak of the COVID-19 pandemic at the end of 2019 led to the closure of the schooling and university education system worldwide in 2020 and again foregrounded the need for multi-modal teaching approaches that ensure that teaching and learning takes place virtually to mitigate any challenges related to face to face tuition. Nearly every university in South Africa was forced to re-evaluate its teaching and learning approaches

with the Department of Higher Education, Science and Technology [2] calling on public universities to produce plans that show how the 2020 academic year would be saved. This was followed by the publication of the 'Quality Assurance Guidelines for Emergency Remote Teaching & Learning and Assessment During the COVID-19 Pandemic' by the Council on Higher Education/CHE [3]. The big question remains: Do South African Institutions of Higher Learning and academics believe or see the value that ICTs bring to education to improve learning and teaching, and are they willing to embrace the use of technology that will transform HE, or is recourse to the use of technology in teaching and learning reactive because of the unforeseen circumstances alluded to above?

The present study sought to examine factors underlying technology acceptance in learning and teaching at a historically disadvantaged university in the Eastern Cape Province of South Africa. The university is a result of a merger of two polytechnic colleges and a university, which operates under a divisional governance model and has four semi-autonomous campuses. The university identifies itself as an impactful and technology infused African university, foregrounding technology as a critical tool for learning and teaching [4]. Although the university introduced blended learning in 2006 as the learning and teaching strategy in the Centre for Learning and Teaching Development Founding Document [5], a very low adoption rate has been witnessed over the years, from less than 20% in 2014 to 48% in 2019 [4]. The Centre for Learning and Teaching Development as the academic development support center in the university is responsible for capacity building of lecturers in integrating information communication technology in learning and teaching in the university. It seems several academics are still far more comfortable with the traditional face to face way of teaching. Evidence shows that more than 75% of the students admitted in any particular year have never had any exposure to learning using technology [6]. This has been exacerbated by the COVID-19 pandemic that struck the nation and the world during the 2020 academic year forcing the university to introduce emergency remote teaching and learning. Given the cultural and contextual challenges identified above, it is imperative that research be conducted to examine the factors underlying acceptance of technology for teaching and learning by university lecturers. This will assist the university to design interventions that will increase such acceptance.

### 1.1. Technology Acceptance in Learning and Teaching in Higher Education
#### 1.1.1. Benefits of ICT Use for Teaching and Learning

The urge to use technology has generally not been embraced with the ease that would have been expected despite the widely reported benefits of integrating information communication technologies in teaching in higher education. Blended learning reduces online transactional distance, increases the interaction between teachers and their students and offers flexibility [7]. This is corroborated by [8] who argue that under ideal conditions technology has promoted flexibility in the place and time to study, accessibility of different teaching and learning resources, personalised ways of teaching and learning and readiness for future digital demands. Similarly, in a study by [9] the teachers described positive experiences regarding independence of place, time and the possibility of individualising the learning environment when using e-learning. [10] argue that as e-learning is not time-bound or static, it has helped the students to access the material from anywhere and at any time. Teachers may develop, improve, and check the learning contents anytime. In South Africa, where this particular study is located, a study by [11] concluded that e-learning provided students with opportunities to manage their own task in their own time which therefore took personal learning to a whole new level. Furthermore, they argue that time and location limit students considerably while [12] avers that the use of e-learning allows lecturers access to a wide range of students anytime and anywhere. The significance of e-learning in mitigating the constraints of time and space is also corroborated by [13,14].

### 1.1.2. Teacher Beliefs and Pedagogical Use of ICT

While the literature is abounded with several benefits for integrating information communication technologies in learning and teaching, acceptance of technology should not be taken as a given as teacher beliefs on the use of technology can have an impact of technology acceptance in the higher education sector. Hew and Brush [15] noted that the challenge associated with technology acceptance comprises not only specific technology usage knowledge but also lack of technology-based pedagogical information. Rasheed, et al. [7] indicated that skepticism about the effectiveness of online instruction in improving learning is one of the reported negative perceptions and beliefs from blended learning teachers regarding using technology for teaching in the literature. In the same vein, Pan [16] reported that previous studies have highlighted that students' beliefs on the utility of technology influenced attitude toward technology use implying that both teacher and student beliefs can affect technology acceptance. Sometimes beliefs may not necessarily only be about the technology but may also be because of a group's culture, norms, and direct influences with respect to use of an educational technology. Kemp, et al. [17] suggested that how one will be perceived by others as a result of using the technology and the degree to which use of the educational technology will augment the esteem or image of the user within a social group may influence technology acceptance.

Belief in the pedagogical value of using technology in enhancing learning may also have a bearing on whether lecturers adopt technology in their teaching. A study by [9] discovered some barriers amongst many teachers in the use of technology such as the lack of direct, personal interaction, which they found unsettling and frustrating in using technology in their teaching and learning. This is in line with the assertion by [18] that failure to examine teachers pedagogical beliefs would lead to limited understanding of the factors of militating against incorporating ICT in classroom teaching. A study by [19] confirms that teachers whose pedagogical approaches are aligned to constructivist beliefs and learner-centred strategies are likely to incorporate ICT in their classroom instruction easily. In the same vein, a study in [17] confirmed that some teachers' beliefs about their inability to use ICT for teaching and learning made them feel insecure resulting in feelings that that ICT was difficult to use for teaching.

Models that attempt to theorize technology acceptance which can apply to the higher education sector are abound in the literature, among them the Theory of Reasoned Action [20] Theory of Planned Behavior [21], Technology Acceptance Model [22] and the Unified Theory of Acceptance and Use of Technology [23]. The study is premised on the Technology Acceptance Model.

### 1.2. The Technology Acceptance Model

This study is premised on Davis [22]'s Technology Acceptance Model (TAM), as an analytical framework for determining factors which influence acceptance of technology in teaching and learning environments. TAM adapts and makes use of the Theory of Reasoned Action [20,24].

The Theory of Reasoned Action (TRA), a model wisely used in social psychology studies [25,26] postulates that an individual's attitude toward behavior is influenced by his/her beliefs [27]. Building on TRA, TAM specifically focused on analyzing "users' willingness to accept and use new technology or media in the field of information system management [27]. "The two most important individual beliefs about using information technology according to TAM are Perceived Usefulness (PU) and Perceived Ease of Use (PEOU) that are able to explain individual's Intention to Use (IU) the technology [28]. Perceived Usefulness is defined as the potential user's subjective likelihood that the use of a certain system will improve his/her action and Perceived Ease of Use refers to the degree to which the potential user expects the target system to be effortless in [15,22,28,29].

"An individual's salient beliefs about a system (perceived usefulness and perceived ease of use) determine his/her attitude towards using the given system [29]". Therefore, as Taherdoost [30] shows, recognition and realization of the needs and factors that drive users'

acceptance or rejection of technologies at the introduction stage would be helpful so that they are taken into account during the development phase. Figure 1 depicts the original TAM model.

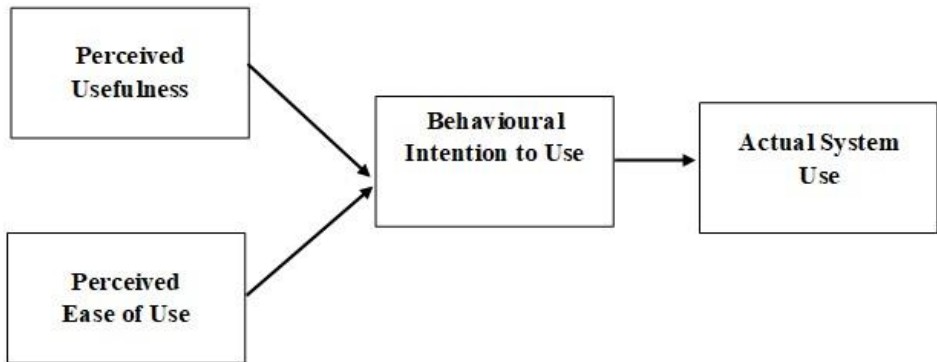

**Figure 1.** Original Technology Acceptance Model.

Through further development, the TAM model was refined to TAM II through provision of more detailed explanations for the reasons users found a given system useful at three points in time: pre-implementation, one-month post-implementation and three-month post implementation [31]. The four major variables of Perceived Usefulness (PU), Perceived Ease of Use (PEOU), Behavioral Intention (BI) and Actual System Usage remain. According to Lee, et al. [32] through synthesis of previous efforts, and reflection on the need for the model's elaboration, Venkatesh and Davis [31] defined the external variables of PU, such as social influence and cognitive instruments which include job relevance, quality, and result demonstrability while Venkatesh [31] provided the external variables of PEOU, such as, "anchor (computer self-efficacy, perceptions of external control, computer anxiety, and computer playfulness) and adjustments (perceived enjoyment and objective usability). Computer self- efficacy, referred by some authors as technological self-efficacy [16], technological pedagogical knowledge (TPK) self-efficacy [8] and digital literacy [33] refers to all those skills, attitudes and knowledge required by teachers in a digitalized world. It also refers to the belief in one's capability to organise and execute internet-related actions required to accomplish assigned tasks [34]. If university teachers have strong ICT-related knowledge, they will be able to overcome ICT related barriers and thus successfully incorporate technology into their teaching practice [33,35]. While computer efficacy significantly predicts continuance intention in e-learning among university lecturers [36,37] it also significantly affects students' behavioural preferences to use technological tools for learning [16,38,39]. This calls for adequate capacity building to build computer self-efficacy among lecturers and students alike in universities.

A study by Lee, et al. [32] that traced TAM's history, investigated its findings, to determine its future trajectory through a review of one hundred and one articles published by leading Information Systems journals and conferences over an eighteen year period concluded that it had been, "elaborated by researchers, resolving its limitations, incorporating other theoretical models or introducing new external variables, and being applied to different environments, systems, tasks, and subjects. It is against this background that the original TAM model is used together with its subsequent refinements in this study to examine technology acceptance among academic staff.

## 2. Materials and Methods

An explanatory sequential mixed methods paradigm approach was selected for this research study to examine technology acceptance for e-learning among academics. The explanatory-sequential approach which is a chronological approach is used when the researcher is interested in following up the quantitative results with qualitative data [40]. The sequential mixed-method (incorporating collection of both quantitative and qualitative data with quantitative data collected first followed by collection of qualitative data) was used in

order to triangulate the data, as well as to solicit rich data from respondents [41]. While the mixed methods approach was adopted for purposes of triangulation of data, the predominant approach used was the qualitative approach through the open-ended questionnaire. Quantitative data was used mainly to help construct the qualitative questions.

### 2.1. Population and Sampling

The Learning and Teaching with Technology (LTwT) Unit in the university periodically conducts e-learning workshops and related training on the integration of information communication technologies in learning and teaching. Purposive sampling was used to select participants for the study. Purposive sampling involves identifying and selecting individuals or groups of individuals that are especially knowledgeable about or experienced with a phenomenon of interest [42]. The phenomenon of interest in this case was lecturer integration of information communication technologies in learning and teaching. Records from LTwT indicated that one hundred and three lecturers had attended these workshops in the period under review. The 103 lecturers were contacted through emails with an explanatory note on the purpose of the study requesting them to indicate if they would be willing to participate in the study. A total of 50 lecturers expressed willingness to participate and constituted the sample for the study. Fifty lecturers were considered adequate as the primary purpose of the study was not generalisation but to identify the factors underlying acceptance of technology for teaching and learning at the chosen university to assist the university to design interventions that would increase such acceptance. Notwithstanding the sample size however, the findings which are corroborated in the literature appear generalisable.

### 2.2. Data Collection

Two sets of online questionnaires, one structured with pre-coded questions and one with open ended questions were sent to lecturers through an online link using university emails and the WhatsApp platform for them to complete. Their contact details were readily available through the email addresses and WhatsApp numbers they had provided in attendance registers during the training. Validity and reliability were ensured through content validity and inter-rater reliability [43] where experts in the Learning and Teaching with Technology Unit (LTwT) were asked to complete the questionnaires and give their opinion about whether the questionnaires captured the topic under investigation effectively and whether or not there were any confusing questions. Based on feedback from the LTwT unit, questions were modified accordingly. The two questionnaires were then converted into 'google docs' and the links emailed to all selected participants. In addition, the google docs links were sent through their WhatsApp platform using their cellphone numbers. A total of 42 questionnaires were received out of the total number of 50 questionnaires sent out. Using the sequential mixed methods approach, the pre-coded questionnaire was sent to respondents first and based on the preliminary analysis of responses, the open-ended questionnaire which had already been designed was amended where necessary to probe on issues emerging from the pre-coded questionnaire.

Building on the assertion that the belief of the person towards a system may be influenced by other factors referred to as external variables [15] the pre-coded questionnaire was designed to solicit information from participants on external factors to the system itself, but which could impact acceptance of the technology, such as ease of internet access, device ownership, availability of and ease of access to system technical support. Rather than use the traditional Likert scale, with items ranging from strongly disagree to strongly agree, actual variables were used as coded responses. The variables used are indicated as a key in the results section on the Figures for the pre-coded questions asked. Taking a cue from previous TAM research [26,29,41] the following constructs; Attitude Towards Using (AU), Perceived Usefulness (PU), and Perceived Ease of Use (PEU) were used in constructing the second open ended questionnaire which sought specifically to examine the determinants of technology acceptance at the university under study. Questions under

'Attitude towards using' sought lecturer views on the use of technology for teaching and learning and lecturer likelihood to use WiSeUp (or any other technology integration tool) for learning and teaching if given freedom to opt out. Under the TAM category of perceived usefulness questions solicited information on reasons why academics used the WiSeUp Learning Management System and lecturer views on suitability of the use of WiSeUp for interaction with students both in an out of class. Questions measuring perceived ease of use gathered data on lecturer access to devices for integration of technology in teaching and learning, ease of access to assistance with challenges associated with using the WiSeUp learning management system and computer literacy/competence skills that impeded lecturer effective use of WiSeUp.

### 2.3. Ethical Considerations

To ensure informed consent an explanatory letter was sent to all participants explaining the purpose of the study prior to commencement. After agreeing to participate, participants then signed consent to participate forms. To ensure anonymity and confidentiality, although the link to the two questionnaires was sent through emails and the WhatsApp platform, participants responded on 'google docs' through the links and this made their identities anonymous. Emails and the WhatsApp platform were thus used only as points of contact and not as points of response to questionnaires. All data was reported as aggregated group data without any reference to individual participant identities.

### 2.4. Data Analysis

As noted under data collection, both quantitative and qualitative data were collected using two sets of online questionnaires one structured with pre-coded questions and one semi-structured with open ended questions completed by 42 lecturers. Data from the pre-coded questionnaire were analysed through the descriptive statistical approach where the raw data was organized and summarized by use of graphical representation [44] using the Statistical Package for the Social Science (SPSS). This was followed by analysis where inferences, interpretation and conclusions were drawn from the quantitative data. The qualitative data from the semi-structured questionnaire was analysed through thematic analysis. Thematic analysis (TA), is a method for systematically identifying, analyzing, organizing, describing and reporting patterns of meaning (themes) found within a data set [45,46]. The thematic analysis involved an idiographic process that started with an iterative and detailed examination of all the individual responses several times for each question and identifying and coding emerging patterns and themes. Open-coding, axial-coding and selective-coding techniques to identify similarities and differences as well as contradictions was done [47]. Through inductive analysis [48], recurring patterns and common themes were identified. Glaser and Strauss [48], developed this approach that has been used widely in qualitative and mixed methods research studies. This approach enables participants' themes to appear from data rather than pushing the data into pre-existing categories.

### 3. Results

These results are presented according to the three TAM categories of Attitude Towards Using (AU), Perceived Usefulness (PU), and Perceived Ease of Use (PEU). Under Attitude Towards Using, lecturer views on the use of technology for teaching and learning and results on lecturer likelihood to use WiSeUp (or any other technology integration tool) for learning and teaching if given freedom to opt out are presented. Perceived Usefulness is presented under the following subheadings: Reasons why academics use the WiSeUp Learner Management System, how the use of WiSeUp affects lecturer productivity and lecturer views on suitability of the use of WiSeUp for interaction with students both in and out of class. Under the category of Perceived Ease of Use results are presented under the subheadings; Lecturer access to devices for integration of technology in teaching and learning, computer literacy/competence skills that impede lecturer effective use of WiSeUp

and ease of access to assistance with challenges associated with using the WiSeUp Learner Management System.

*3.1. Attitude towards Using (AU)*

- Lecturer Views on the use of technology for teaching and learning

To gauge participants' attitudes towards the use of technology for teaching and learning a question soliciting their views on the use of technology for teaching and learning was paused. Thirty-five out of the forty-two participants view the use of technology in positive light seeing it as necessary in the wake of the COVID-19 outbreak where students had to leave campus. Technology is therefore seen as an opportunity to ensure that students could continue to learn remotely. The context of the 4th industrial revolution was also given as one reason lecturers felt technology should be embraced in order not to be left behind. The following excerpts are sample responses in this regard:

> *I think it will really assist since we were faced with COVID-19 and it will help to be aligned with other institutions, so that we won't be left behind in this new era of 4th Industrial revolution.*

> *As we approach 4th industrial revolution, technology is becoming a core competency in offering fast and efficient services.*

> *It a necessary tool for teaching and learning in this day and age.*

There was also a group of academics who felt apprehensive regarding the use of technology citing capacity to use the technology, fearing that students from disadvantaged backgrounds might be left behind. Some questioned the timing of accelerating the use of technology during times of crises (in this case under COVID-19) arguing that such interventions should be introduced under normal circumstances. There was anxiety around the issue of training as shown in these sample responses:

> *My view is that we need lots and lots of training for us to use technology for T&L.*

> *We must be cognizant of trying to introduce new ways of teaching and learning during the time of crisis like this. New ways of doing things must be introduced and be mastered while things are still normal.*

> *It is convenient during this time of the Pandemic; however, it is less convenient for students who are in the most rural areas.*

> *4IR requires of us to use technology. It is good but not fair to students.*

A related question sought to ascertain participant views on fear of being de-skilled with the introduction of the technology. Most of the participants (35 out of 42) had no underlying fears at all regarding the use of the technology indicating their willingness to learn where need be.

> *No, I don't fear using technology, as humans we are always learning new things in life, it is not alien that change is inevitable and systems are always evolving, and one needs to always be willing to adapt and be trained on using new systems.*

> *Not really, a necessary ongoing learning process for personal development as well.*

> *I don't share the same sentiment especially if there is training taking place that will equip everyone to use technology.*

> *No. We are living in an era that is becoming more digital by the day and thus is it necessary to adapt to the world of 4th industrial revolution.*

> *No. There are academics who use WiSeUp. There is no doubt that it is challenging but it also gives academics several functions to explore for teaching.*

Five out of the forty-two participants did indeed fear that technology would deskill them as they felt they did not have the craft literacy and craft competency to embrace and

use the technology. There were some, who although did not fear introduction of technology in teaching and learning, nevertheless saw training as a precondition before e-learning could be rolled out.

> *Yes, many people are not trained on WiSeUp up and this makes it difficult to use it.*
>
> *For me I see as a good system, but again thorough training must be provided.*
>
> *Yes, I agree but with proper training not a problem.*
>
> *For me I see it as a good system, but again thorough training must be provided.*
>
> *I don't share the same sentiment especially if there is training taking place that will equip everyone to use technology.*

- Lecturer likelihood to use WiSeUp (or any other technology integration tool) for learning and teaching if given freedom to opt out

Further probed to indicate what they would do if they were left to decide on whether to use WiSeUp (or any other technology integration tool) and there was no compulsion from the university on the use of technology in teaching and learning, all the forty-two participants would opt to integrate technology in teaching and learning anyway. Cited reasons included the fact that students tended to be more actively engaged when learning online when compared to face to face tuition. The need to ensure that the university's students would compete equally in the technologically biased global economy was also cited as a reason for opting for technology even if this was not legislated in the university. The need to ensure learning continued actively beyond the classroom was another reason participant would opt for technology integration out of their own volition.

> *Use of technology is good. I would choose technology over any other way. It forces one to learn especially if monitored and eases the work of the lectures.*
>
> *To be quite honest, face to face teaching should be necessary only if there are specific topics that need both the lecturer and students. Most students come to class because of the "attendance register" and do not quite engage so much in class compared to when we're discussing something on an online platform e.g., WiSeUp or WhatsApp.*
>
> *I think it would be unfair for students in our institution if we do not use technology, because the quality of students we will be graduating will not have the competencies and skills required by organizations.*

There was however caution not to abandon the traditional methods implying a blended learning approach to accommodate those students who could be late adopters. Coupled with this again the need for training was given as a pre-condition for voluntarily deciding on whether to use the technology.

> *I would decide on using WiSeUp as it makes teaching and learning much accessible and easier. But I would not abandon the traditional ways of teaching because some students are late adaptors.*
>
> *Will choose WiSeUp and other technology integration with training or assistance back up.*
>
> *I would use WiseUp if properly trained.*

### 3.2. Perceived Usefulness

- Reason why Academics use the WiSeUp Learning Management System

To solicit lecturer responses on the perceived usefulness of the university's learner management system, lecturers were asked to explain why they used the system. The need to reach as many students as possible within a short space of time, technology's ability to allow lecturers to work remotely and reach their students, and its ease of access wherever students are beyond the classroom were some of the justifications given for using technology in teaching and learning. The fact that once uploaded, material remained on the system and students, including those who might have missed the lecturers could access material at their convenience.

It's easier to manage and you can see who is participating or not based on the design of the system.

*To promote effective teaching and learning outside the classroom and easy access to all irrespective of where you are.*

*For its convenience. Firstly, some information uploaded will always be accessible for the rest of their academic year.*

*Secondly if I'm not able to meet students physically I can always upload notes or work on WiseUp.*

*We are living in an era that is becoming more digital by the day and thus is it necessary to adapt to the world of 4th industrial revolution.*

From those who did not derive satisfaction from using WiSeUp, the main reason given was the issue of challenges with connectivity:

*The challenge of accessibility to technology makes me disinterested.*

*I have students in remote areas with internet access challenges.*

- Lecturer views on suitability of the use of WiSeUp for interaction with students both in and out of class

A probing question on the usefulness of the learner management system regarding the issue of student lecturer interaction in an online environment was included in the open-ended questionnaire. The fact the LMS enabled teaching and learning to continue beyond the classroom, enabled students to prepare for face-to-face lectures in advance resulting in greater engagement in class, quicker response rate from students and the opportunities for offering continuous feedback to student students at their convenience were among the reasons cited under perceived usefulness. The opportunities offered by the LMS to help mitigate teaching and learning disruption during times of crises such the national lockdown promulgated in 2020 was also cited by 29 out of 40.

*It is good because I get quick responses from my students before and after class.*

*I find it to be very relevant as teaching and learning continues outside class.*

*Students nowadays use smartphones, tablets, and laptops. The university has implemented WiFi services across the university premises. Keeping connected with students both in class & outside class makes learning easier.*

*WiSeUp is much relevant because students who have managed to interact with the content on WiSeUp are usually coming for lectures prepared.*

*Promotes continuous feedback and assessment for student performance improvements.*

Eleven of the forty-two participants appeared pessimistic on the issue of usefulness of the LMS. Some argued that it was effective out of class but in class, while one some indicted that it was useless as students were inactive on the system with very minimal participation. The issue of training was again brought up as a condition before the system could be found useful.

*Very inactive from students.*

*Effective out of class not so much in class.*

*Out of class-I can send a link for submissions and restrict the duration time and create discussions platform but I have not used the discussion forum platform as yet with my students, still need training.*

*Very minimal.*

### 3.3. Perceived Ease of Use (PEOU)

- Lecturer access to devices for integration of technology in teaching and learning

Under the pre-coded questions there was a question that sought to ascertain provision of resources by the university for ease of use of the learner management system. 90% of

the participants used their own laptops and accessed internet facilities online at their own homes and not at the university. 13% of the participants did not have private internet accessibility at their homes meaning they could not work on the LMS at home. This foregrounds the need for data provision for academics beyond the university precincts.

Asked on a pre-coded question on whether they had access to reliable internet at work, it was concerning to note that over half of the participants (54.5%) as shown in Figure 2 either often or very often had challenges with internet access. The prerequisite for e-learning is reliable internet connectivity to be able to use the learner management system and unreliable internet has a negative impact on ease of use

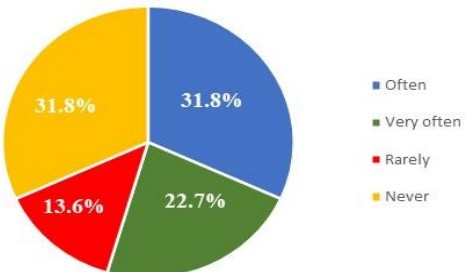

**Figure 2.** Reliability of internet access for teaching and learning.

- Computer literacy/competence skills that impede lecturer effective use of WiSeUp

To follow up on specifics about perceived ease of use, participants were asked to enumerate computer literacy/competence skills they found to impede effective use of WiSeUp. Twenty-five of the participants did not experience any impediments while 17 of the 42 participants have grey areas they felt could be mitigated through training; The main training need given was the need for Microsoft excel training:

*Preparation of online assessments.*

*Basic Computer Literacy.*

*Microsoft Excel for assessments.*

*Loading all work to monitor learner progress.*

- Ease of access to assistance with challenges associated with using the WISeUp learner Management system

Asked what mode they used to seek technical support, (Figure 2), 54.5% indicated that they relied on email communication, 18.2 percent on telephone support and technicians on site respectively and 9.1% on call centre support.

A probing pre-coded question on satisfaction with time normally taken to receive the support requested after logging a query was worrying to note that only 18.2% (Figure 3) were receiving immediate support upon request.

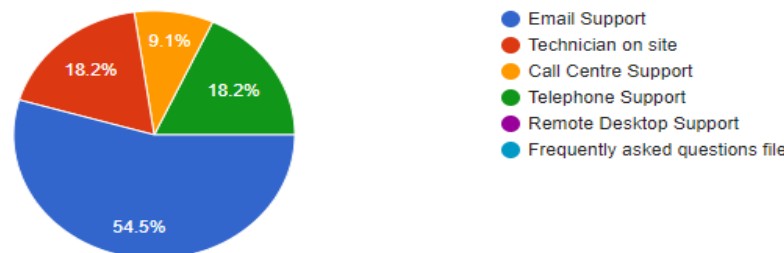

**Figure 3.** Method used to get technical support.

Regarding time taken to receive support, eleven of the participants felt it was challenging to receive support when they experienced challenges with using the WiSeUp Learning Management System while 31 were in the affirmative and one sat on the fence arguing that

it depended on circumstances at the time. As shown in Figure 4, it is concerning that 40.9% of the participants never received the required support and 31.8% only received support after following up several times when they logged requests with the technical support department. Only 18.2% received support timeously. When lecturers do not receive the required support their perceptions of ease of use of the technology declines leading to rejection of technology. For these who were positive reasons given included the fact that the e-learning specialists were readily available when needed:

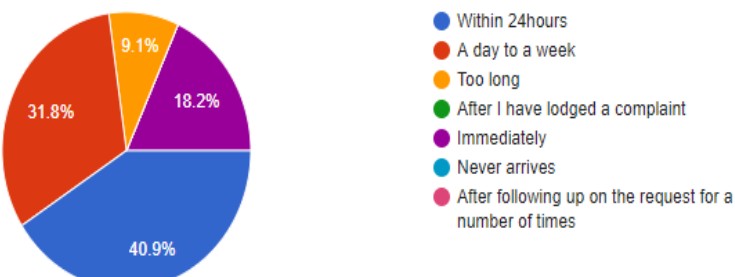

**Figure 4.** Time taken to get support.

> *It's easy. The e-teaching and learning specialist is very diligent.*
>
> *It is easy. CLTD are always willing to go above and beyond to assist with all the challenges that I face.*
>
> *Not difficult. Staff is always ready to help.*
>
> *Not difficult at all for me. Key colleagues and CLTD are on speed dial and always ready to assist even after hours.*

The fact that while some felt support was adequate others felt there was no support could probably be due to the divisional model (where individual campuses are semi-autonomous) where support on some campuses could have been adequate while not adequate at other campuses.

> *Difficult to get any kind of help related to WiseUp.*
>
> *It easy to get assistance but sometimes I run out of data.*
>
> *Difficult because of staff shortage and time constraints.*
>
> *It is difficult not enough support.*

## 4. Discussion

The contiguous approach to integration was used in the preceding section on presentation of results, which comprised the presentation of findings with the qualitative and quantitative findings reported in separate subdivisions [40]. In this discussion of results section, the weaving approach to integration was used where both qualitative and quantitative results are discussed simultaneously on a theme-by-theme basis [40] using the three TAM concepts of Attitude Towards Use, Perceived Ease of Use and Perceived Usefulness.

*4.1. Attitude towards Using (AU)*

- Lecturer views on the use of technology for teaching and learning

As Surendran [49] shows the attitude towards use is concerned with user's evaluation of the desirability of employing a particular e-learning system and is a measure of the likelihood of the person using the system. Regarding attitudes towards using, qualitative results of the study show general positive attitudes towards the use of technology for teaching and learning. This is corroborated in the quantitative data where thirty-five out of the forty-two participants view the use of technology in positive light seeing it as necessary in the wake of the COVID-19 outbreak where students had to leave campus. Technology is therefore seen as an opportunity to ensure that students could continue to learn remotely.

Thus, users' mental assessment of the match between important goals at work (successfully completing the academic year in this instance) and the consequences of performing job tasks using the system (using technology to ensure students learn remotely from home due to COVID-19 restrictions) serve as a basis for forming perceptions regarding the usefulness of the system [15,28,31]. Hoong, Thi and Lin [28] further alluded that individuals rely on the fit between their job and the performance outcomes of using the system before concluding on usefulness of the system. They argue that if the system does not produce any desirable output to enhance individual performance, the user acceptance rate is likely to drop. The concern about teacher fear in the use of ICT is confirmed by [50], who indicated that even though some teachers believe in constructivist pedagogy, they are still reluctant to use technology because of various constraints such as lack of adequate time to design lessons for online delivery, insufficient specialized help, absence of physical contact with students as well as challenges related to internet connectivity. [34] advise that factors that should be taken into consideration, besides the teaching process and instructional content, are e-competencies of students and educators, as well as the attitudes toward this mode of learning and the usability of the system. In this study, the opportunities provided by the e-learning system to save the academic year through moving to online teaching and learning to mitigate the effects of COVID-19 in face-to-face tuition seems to have cultivated positive attitudes towards use.

Five of the participants in this study however, though few, as shown in the results, felt apprehensive and therefore had negative attitudes towards the use of technology citing capacity to use the technology. To mitigate this challenge of poor technology acceptance, Ibrahim and Nat [51] call for provision of professional development programmes specifically for pedagogical and technological skills. A study [52] concludes that the level of competence regarding the use of technological tools can be improved in the need to understand that the development of virtual teaching also entails the need to develop and enhance the competence part linked to interaction and communication with students. Similarly, [33] argue that more ICT teacher training means better training conditions for students and recommend that teachers be trained in both technological and pedagogical areas in order to develop digital teaching skills.

Some lecturers in this study questioned the timing of accelerating the use of technology during times of crises (in this case under COVID-19) arguing that such interventions should be introduced under normal circumstances. This finding corroborates findings by Johnson et al. [53], which showed that teachers most of the time viewed technology as an imposition. People need to feel at ease when implementing interventions as opposed to feelings of pressure and compulsion. As Rossouw and Alexander [54] pointed out, people also need to feel they have some individual control over change as group needs, organizational needs and individual needs are not synonymous and should be addressed differently.

- Computer literacy/competence skills that impede lecturer effective use of WiSeUp resulting in lecturers being de-skilled with the introduction of the technology

Fear of the unknown can sometimes affect attitudes towards emerging innovations. While many of the participants (35 out of 42) as shown in the quantitative results had no underlying fears at all regarding the use of the technology indicating their willingness to learn where need be, five out of the forty-two participants as shown the qualitative data felt that technology would deskill them as they felt they did not have the craft literacy and craft competency to embrace and use the technology. The results assert findings by Portz et al. [26], who found in their study that perceived ease of use was impacted by participants' level of computer anxiety and computer self-efficacy. Thus, the absence of technological literacy slows blended learning applications among lecturers, and frequent interaction with technology encourages the intention to blend among instructors [51]. Ibrahim and Nat [51] further argued that capacity building in relation to training is the most critical support that a lecturer can tap from the institution. Reporting from China, [55] concluded based on the data gathered in their study that factors affecting Chinese English teachers' online

teaching provide suggestions for policymakers and teacher professional development, such as improvement in technical support, and provision of technology training.

One way of enhancing this capacity building would be through encouraging the formation of communities of practice so academics can share their practice and support each other in integrating information communication technologies in teaching. Communities of practice are groups of people who are willing to spend time together to share information, insight and advice where members ponder common issues, explore ideas and act as sounding boards for each other's ideas [45,56]. Members in a community of practice engage in joint activities and discussions, help each other and share information and build relationships that enable them to learn from each other [57–60]

- Lecturer likelihood to use WiSeUp out of own (or any other technology integration tool) for learning and teaching if given freedom to opt out

Of consensus in the results (qualitative data) is the assertion by all lectures in the study that they would continue to use WiSeUp (or any other technology integration tool) even if there was no compulsion from the university on the use of technology in teaching and learning. Cited reasons included the fact that students tended to be more actively engaged when learning online when compared to face to face tuition. The belief that technology promoted student engagement was thus a motivator that led to justification for use of the technology as it was believed this would lead to active learning in the classroom. A study by Johnson [53], shows that teachers attitudes and beliefs in the use of technology are crucial factors that determine the role and effectiveness of technology in the classroom. The need to ensure that the university's students would compete equally in the technologically biased global economy was also cited as a reason for opting for technology even if this was not legislated in the university. The need to ensure learning continued actively beyond the classroom was another reason lecturers would opt for technology integration out of their own volition. The results demonstrate that participants have assessed the potential that technology has on their work and resolved that the introduced system responds to both their current needs and those of their students. As Hoong, Thi and Lin [28], conclude, individuals will assess whether the technology constitutes a threat or an opportunity and how it can adapt into their daily tasks by changing their working behavior. In this instance the participants have resolved to use the technology out of their own free will.

*4.2. Perceived Usefulness (PU)*

- Lecturer views on suitability of the use of WiSeUp for interaction with students both in and out of class

In line with the Perceived Usefulness (PU) tenet of the Technology Acceptance Model (TAM), results of this study have demonstrated that indeed the adoption of technology in learning teaching at the university under study depended on the extent to which it was seen as relevant and useful in the learning and teaching process. Hoong, Thi and Lin [28], argued that perceived usefulness (PU) is characterised as how much individuals trust that utilising a specific tool would improve their performance and, "is the key determinant that emphatically influences users' convictions and expectation to utilize the innovation." With regards to the perceived usefulness of the university's learning management system (WiSeUp), lecturers cited the need to reach as many students as possible within a short space of time, technology's ability to allow lecturers to work remotely and reach their students and its ease of access wherever students were beyond the classroom as factors that would enhance their performance under the COVID-19 circumstances. This is, firstly, because the easier a user feels it is to use a new technology or service, the more useful lecturers perceive it to be [61] and, secondly, because the time and effort required to use online educational services are reduced, thus making the service more convenient [55]. The fact that once uploaded, material remained on the system and students, including those who might have missed the lecturers could access material at their convenience constituted justification for usefulness of the WiSeUp. This is in line with a study by Ertmer et al. [62]

who discovered that teachers are able to enact technology integration practices that are closely aligned with their beliefs.

Further perceived usefulness, from the results can be seen in the manner lecturers saw the use of WiSeUp positively affecting their productivity and effectiveness. The flexible accessibility of the system by students and the efficacy it brought, saved valuable time. The ability to send additional links to students after online teaching, the workload reduction in assessment, specifically for Multiple Choice Questions (MCQs) were lauded. WiSeUp also ensured that even if a lecturer had to be away, for example to attend a meeting, they could upload the lessons online and learning would continue in their absence. As e-learning is not time-bound or static, it helped the students to access the material from anywhere and at any time (Patra, Sundaray and Mahapatra 2021) What emerges here is the perceived impact of technology on productivity. The attitude of an individual is not the only factor that determines his/her use of new technology, as the impact the tool or system will have on his/her performance is also significant [22,41]. Literature shows that when teachers believe that technology connects directly with their specific content areas and/or grade levels, as well as allowing them to more readily meet their classroom goals they likely have a tendency to use it frequently [63,64].

### 4.3. Perceived Ease of Use (PEOU)

- Lecturer access to devices for integration of technology in teaching and learning

Perceived ease of use (PEOU) is defined as the degree to which the prospective user expects the intervention or system being introduced to be free of effort [49]. Results of this study show that the amount of effort or resources that lecturers must find on their own to use a system has a dent on perceived ease of use of the system. The fact that participants used their own laptops and internet facilities to be online at their own homes as shown in the quantitative data and not at the university meant that when they ran out of resources to purchase such accessibility, they could not work on WiSeUp at home. In addition, quantitative findings of the study revealed that half of the participants had no access to reliable internet at work. The prerequisite for e-learning is reliable internet connectivity to be able to use the learning management system and unreliable internet has a negative impact on ease of use. Some studies have identified issues such as lack of lecturer preparation for online learning, constraints on learning facilities that are not fully ready and complete for students and technical obstacles such as the internet network that many students complain about during online learning [17,18]. Ibrahim and Nat [51] contend that lack of access to appropriate hardware and software can slow and suppress the highest motivation. Participants in focus group discussions in a study by Chigona [65] asked for more digital resources such as reliable software and Wi-Fi and believed that making available such requisite resources could be the answer to educators' adoption and use of connected classrooms effectively. It is indisputably disappointing for the educators when they do not have adequate resources to implement their ideas or work with the system [65].

### 4.4. Ease of Access to Assistance with Challenges Associated with Using the WISeUp Learner Management System (Training and Technical)

Tied to the issue of internet connectivity, the findings revealed that the extent to which lecturers feel they are comfortable to navigate the WiSeUp learning management system had a bearing on perceived ease of use. While it is laudable as the quantitative data revealed, that 25 of the 42 participants did not experience any impediments, the 17 participants who indicated need for training will need to be prioritized to improve their perceived ease of use. In the literature, studies show that if teachers do not have necessary competencies in using technology, they are unlikely to explore new possibilities to utilize technology compared to those who have the knowledge and skills in the use of technology frequently [33,56,64,66]. A study by Nair and Das [67] revealed that teachers would find the information technology (IT) tools more useful and will have a positive attitude towards

integration of technology in teaching if through adequate training they are made more proficient in using such tools.

Further to the issue of knowledge of the system, the provision of the requisite technical support to navigate the system when need arose and the extent of satisfaction with such support was found to influence perceived ease of use. As shown in the quantitative results only 18% of the participants normally received support requested immediately after logging a query with 28% indicating that it was challenging to receive support at all when they experienced challenges with using the WiSeUp Learning Management System. Rossouw and Alexander [54], suggested that users experience the system as technology and if the system functions without any problems, then the technology is not a problem. For those who are not technology-savvy, time and effort must be invested to perform these operations in addition to ensuring that the pedagogical aspects of the course are managed effectively and lack of support creates stress and increases teachers' perceptions of complexity of the technology system [55]. In the same vein, Mbodila Ndebele and Muhandji [68] confirm that the integration of new technology for the purpose of teaching and learning depends on level of support and guidance that is provided to both teachers and students in the use of the new technology. A different study by Hu and Garimalla [69] confirms that professional development such as training to support teachers in the use of technology is one way to promote technology adoption. The differential views in satisfaction with support in this study could be attributed to the divisional model in the university (where campuses are semi-autonomous) where support is provided per campus and where support on some campuses could have been adequate while not adequate at other campuses.

## 5. Conclusions

In conclusion, the integration of technology in teaching and learning has seen increased focus in the higher education systems around the world and continues to be a significant area of research today. Most higher education institutions around the world and in South Africa have integrated various learning management systems (LMSs) to deliver teaching and learning in a blended fashion. However, there are still challenges of slow adoption amongst academics in many institutions. The results of this study show that most academic staff still believe and see the value that ICTs bring in their teaching and learning practices. In addition, they are aware that technology use in education improves learning and teaching, and they are willing to embrace the use of technology to improve their practices. However, there is a need for the HEI to provide requisite training, support, resources and tools of trade to enable lecturers to make continuous use of technology in teaching and learning even beyond COVID-19 pandemic. Based on the above findings, the following recommendations are put forward:

- Intensification of lecturers training in the use of technology for teaching and learning to enable them to embrace it in their teaching practice. This will assist is removing any fear of the unknown and to view technology as tools that enhance the teaching and learning experience.
- The institution needs to put in place support systems for academic staff to empower them to have continuous access to devices and internet connection for technology integration in teaching and learning. Provision of tools of trades such as laptops, data and other equipment will enable them to become effective in their practices through 'ease of use'.
- Establishment of e-learning communities of practise that will allow lecturers to assist each other as well as share best practice in the use of technology for teaching and learning. This communities of best practice will promote collaboration and help increase academic buy-in and acceptance of technology integration in teaching and learning.

**Author Contributions:** Conceptualization, C.N.; methodology, C.N.; formal analysis quantitative data, M.M.; formal analysis qualitative data, C.N. writing—original draft preparation, M.M. writing—

review and editing, M.M. and C.N. All authors have read and agreed to the published version of the manuscript.

**Funding:** The work described in this paper was partially supported from the Directorate of Learning and Teaching, and Department of Information Technology Systems, at the university under study through payment of page fees

**Institutional Review Board Statement:** Ethical review and approval were waived for this study, since individuals were contacted individually in their individual capacity and gave consent in their individual capacity and not in their capacity as members of a particular university. Secondly, the name of the university is not mentioned in the study to maintain the anonymity assured to participants.

**Informed Consent Statement:** As detailed in the sub-section on ethical considerations in this article, informed consent was obtained from all subjects involved in the study.

**Data Availability Statement:** The datasets used/or analysed during the current study are available from the corresponding author on reasonable request.

**Acknowledgments:** The authors acknowledge the students and staff of the university, the Directorate of Learning and Teaching, and Department of Information Technology Systems, Walter Sisulu University who participated in the study and consented to being acknowledged.

**Conflicts of Interest:** The authors declare no conflict of interest. The funders had no role in the design of the study; in the collection, analyses, or interpretation of data; in the writing of the manuscript, or in the decision to publish the results.

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
