# Peer review of "Examining Technology Acceptance in Learning and Teaching at a Historically Disadvantaged University in South Africa through the Technology Acceptance Model"

_education, doi:10.3390/educsci12010054_

Round 1
Reviewer 1 Report
The topic of the manuscript is very relevant. The authors introduce various aspects why multi-modal teaching approaches must be developed. The literature that the authors refer to is relevant and up-to-date. Beliefs related to technology use are reached from the perspective of usability of the product. This section could benefit from some references to the literature related to teacher beliefs, because they may influence in the background, and might connect the topic more closely with the field of education. What kinds of beliefs there are related to pedagogical use of ICT. Also, I would like to see the benefits of ICT use for performance and well-being to be considered. Pedagogical skills are referred to in the discussion, but I would like to see a bit more profound elaboration of this topic. The concept computer self-efficacy should be dealt in more detail. To sum up, the pedagogical aspect in the manuscript should be strengthened, as the use of technology in a pedagogical setting is not just a technical and usability issue.
The use of whatsapp as a research instrument should be considered from anonymity and confidentiality point of view. The research ethics should be considered in a more detailed way. The data analysis is described carefully, and it is easy to follow. The excerpts from the data increase feasibility of the interpretation. The conclusions and suggestions are relevant.
The language of the ms is fluent and easy to read.
Reviewer 2 Report
ABSTRACT: The method he describes is not the one quoted in the article, which refers to a mixed method.There is no need to cite the statistical package used.No significant quantitative data is given. Practical communities are recommended but not explained in the article.
INTRODUCTION. The research question is stated and the objective is specified: to examine the acceptance of technology in learning and teaching.At the end of this introduction this objective is rephrased as: to examine the factors underlying the acceptance of technology in teaching and learning of university teachers in order to design interventions to increase the acceptance of technology in teaching and learning. The aim of the research needs to be unified.The study is contextualised in a historically disadvantaged university in the Eastern Cape province of South Africa. Data is provided but the sources of the data are not correctly cited.The university adopted blended learning in 2009, where - in its strategic plan?The bibliographic citations are not well done, e.g. the 5
THEORETICAL FRAMEWORK. Much has been published on these issues in recent months. There is a lack of bibliographical references such as:
del Arco, I.; Silva, P.; Flores, O. University Teaching in Times of Confinement: The Light and Shadows of Compulsory Online Learning. Sustainability 2021, 13, 375. doi: 10.3390/su13010375
Ramos-Pla, A.; del Arco, I.; Flores Alarcia, Ò. University Professor Training in Times of COVID-19: Analysis of Training Programs and Perception of Impact on Teaching Practices. Educ. Sci. 2021, 11, 684. doi: 10.3390/educsci11110684
Wei, H. C., & Chou, C. (2020). Online learning performance and satisfaction: do perceptions and readiness matter?. Distance Education, 41(1), 48-69.
Kovačević, I.; Labrović, J.A.; Petrović, N.; Kužet, I. Recognizing Predictors of Students’ Emergency Remote Online Learning Satisfaction during COVID-19. Educ. Sci. 2021, 11, 693. https://doi.org/10.3390/educsci11110693
Fernández-Batanero, J.M.; Román-Graván, P.; Montenegro-Rueda, M.; López-Meneses, E.; Fernández-Cerero, J. Digital Teaching Competence in Higher Education: A Systematic Review. Educ. Sci. 2021, 11, 689. https://doi.org/10.3390/educsci11110689
Before going into the TAM model, it would be interesting to take a brief look at these issues based on published studies.
METHODS. The total population is not specified. We are talking about convenience sampling and a resulting sample of 42 informants out of 50 questionnaires sent out. Is this sample significant? This would have to be argued. Two questionnaires were used, one structured with pre-coded questions and one with open questions. Nothing is explained about the validity and reliability of the instruments used and this is important. Were these instruments constructed ad-hoc? Their parts and the items covered by each part should be presented.
Pre-coded questions means that they follow a likert scale? Not clear
RESULTS. The quantitative data analysis is very simple, with percentage inputs. No inferential analysis is done as previously announced.
It is not explained how the analysis of qualitative data is done, if it was previously categorised.
The mixed quantitative-qualitative method. It is not clear which of the two methodologies is predominant, whether they were done simultaneously or not. The mixed method needs to be better explained.
The results are presented according to the three TAM categories of Attitude towards Use (AU), Perceived Usefulness (PU) and Perceived Ease of Use (PEU). However, the different subcategories that appear, which are not the same as those that appear in the discussion section, have not been explained
The triangulation of the results is not clear.
It should be asked whether independent variables such as age, gender, academic qualifications, teaching experience, may affect the results of the dependent variables.
DISCUSSION. In the discussion there are categories and subcategories that are not cited in the results section. No reference is made to authors who have already published in the last year on these topics. In this report I have cited some examples
CONCLUSIONS: The conclusions are well presented and the recommendations are interesting. The implementation of e-learning communities of practice is recommended, but at no point has this strategy been presented, nor has it been argued with other studies on its possible effectiveness.
This article is interesting, I think it should reference other similar studies in higher education. The sample is too small to generalise results, it is more of a case study.
The weakest part is the data analysis, the statistical part (descriptive and inferential) and the triangulation with the qualitative data.
Reviewer 3 Report
I have no comments, the article is fine.
Round 2
Reviewer 2 Report
Hubiera preferido una carta específica dirigida a cada revisor y no una tabla donde cada revisor tiene que seleccionar las respuestas que los autores dan a todos los revisores.
El texto se ha mejorado sustancialmente con nuevas contribuciones de los autores.